# Mortality and associated risk factors of COVID-19 infection in dialysis patients in Qatar: A nationwide cohort study

**Tarek Abdel Latif Ghonimi, Mohamad Mahmood Alkad** , **Essa Abdulla Abuhelaiqa, Muftah M. Othman, Musab Ahmed Elgaali, Rania Abdelaziz M. Ibrahim** , **Shajahan M. Joseph, Hassan Ali Al-Malki, Abdullah Ibrahim Hamad** *

Hamad Medical Corporation, Doha, Qatar

* ahamad9@hamad.qa

**Data Availability Statement:** All relevant data are within the manuscript and its Supporting Information files.

## Abstract

### Context

Patients on maintenance dialysis are more susceptible to COVID-19 and its severe form. We studied the mortality and associated risks of COVID-19 infection in dialysis patients in the state of Qatar.

### Methods

This was an observational, analytical, retrospective, nationwide study. We included all adult patients on maintenance dialysis therapy who tested positive for COVID-19 (PCR assay of the nasopharyngeal swab) during the period from February 1, 2020, to July 19, 2020. Our primary outcome was to study the mortality of COVID-19 in dialysis patients in Qatar and risk factors associated with it. Our secondary objectives were to study incidence and severity of COVID-19 in dialysis patients and comparing outcomes between hemodialysis and peritoneal dialysis patients. Patient demographics and clinical features were collected from a national electronic medical record. Univariate Cox regression analysis was performed to evaluate potential risk factors for mortality in our cohort.

### Results

76 out of 1064 dialysis patients were diagnosed with COVID-19 (age 56±13.6, 56 hemodialysis and 20 peritoneal dialysis, 56 males). During the study period, 7.1% of all dialysis patients contracted COVID-19. Male dialysis patients had double the incidence of COVID-19 than females (9% versus 4.5% respectively; p<0.01). The most common symptoms on presentation were fever (57.9%), cough (56.6%), and shortness of breath (25%). Pneumonia was diagnosed in 72% of dialysis patients with COVID-19. High severity manifested as 25% of patients requiring admission to the intensive care unit, 18.4% had ARDS, 17.1% required mechanical ventilation, and 14.5% required inotropes. The mean length of hospital stay was 19.2 ± -12 days. Mortality due to COVID-19 among our dialysis cohort was 15%. Univariate Cox regression analysis for risk factors associated with COVID-19-related death

**Funding:** The author(s) received no specific funding for this work.

**Competing interests:** All authors have declared that no competing interests exist.

in dialysis patients showed significant increases in risks with age (OR 1.077, CI 95%(1.018–1.139), p = 0.01), CHF and COPD (both same OR 8.974, CI 95% (1.039–77.5), p = 0.046), history of DVT (OR 5.762, CI 95% (1.227–27.057), p = 0.026), Atrial fibrillation (OR 7.285, CI 95%(2.029–26.150), p = 0.002), hypoxia (OR: 16.6; CI 95%(3.574–77.715), p = <0.001), ICU admission (HR30.8, CI 95% (3.9–241.2), p = 0.001), Mechanical ventilation (HR 50.07 CI 95% (6.4–391.2)), p<0.001) and using inotropes(HR 19.17, CI 95% (11.57–718.5), p<0.001). In a multivariate analysis, only ICU admission was found to be significantly associated with death [OR = 32.8 (3.5–305.4), p = 0.002)].

## Conclusion

This is the first study to be conducted at a national level in Qatar exploring COVID-19 in a dialysis population. Dialysis patients had a high incidence of COVID-19 infection and related mortality compared to previous reports of the general population in the state of Qatar (7.1% versus 4% and 15% versus 0.15% respectively). We also observed a strong association between death related to COVID-19 infection in dialysis patients and admission to ICU.

## Introduction

Coronavirus disease 2019 (COVID-19) infection emerged in Wuhan, China in December 2019 and has spread rapidly worldwide [1, 2]. On March 11, 2020, WHO declared COVID-19 a pandemic. COVID-19 is a single-strain RNA virus that typically causes respiratory damage in humans and animals. Severe infections can lead to multisystem disorders. The clinical presentation is highly variable, from an asymptomatic or very mild course (80%) to severe involvement with unilateral or bilateral pneumonia (15%) and a very serious course with acute respiratory distress syndrome requiring ventilatory support in the intensive care unit (ICU) in 3–5% of cases [1, 3]. In severe cases of COVID-19, the immune response can trigger a strong inflammatory reaction accompanied by a cytokine storm that may worsen respiratory symptoms leading to death [1–3]. The mortality rate in the general population ranges from 1.4% to 8% [1, 3], and it increases significantly in patients requiring ICU admission [3].

Patients with kidney disease appear to be at high risk for COVID-19 and its related complications, as most of them are elderly and have multiple comorbidities, and some of them might be taking immunosuppressive drugs to treat an autoimmune disease or a failed kidney allograft [4]. Dialysis patients have additional risk factors, including chronic immune dysfunction, the need to go to the hospital for hemodialysis (HD), and sharing rooms with other patients [5]. Therefore, once infected, dialysis patients become itinerant sources of spreading the infection within this high-risk group. Thus, it can be stated that a dialysis unit is suitable to collect the epidemiology of COVID-19. Reports suggest a more severe disease course in patients with chronic kidney disease [6]; however, outcomes in dialysis patients are still unclear, with earlier small case series suggesting a milder course [7].

The state of Qatar has a total population of 2,723,624 [8], with 878 HD and 186 PD patients. The country has three renal centers; the largest is found in the Hamad General Hospital, which runs four satellite hemodialysis units. The other two centers are in the Al-khor Hospital and Al Wakrah Hospital. All dialysis patients who contracted COVID-19 infection dialyzed in a 6-station mobile dialysis unit until they became COVID-19 PCR negative to limit the spread of the disease.

The purpose of this study is to determine the incidence and mortality of covid-19 infection in patients under dialysis in the state of Qatar.

## Materials and methods

### Study design and population

This was an observational, analytical, retrospective, nationwide study from the state of Qatar. It was done by the Hamad Medical Corporation (HMC), the only healthcare provider for peritoneal and hemodialysis in Qatar. We included all end-stage renal disease (ESRD) patients on maintenance dialysis therapy who tested positive for COVID-19 from February 1, 2020 to July 19, 2020. Patients older than 18 years who had received dialysis for more than 1 month in any of the country's ambulatory dialysis units were enrolled in the study. The study protocol was approved by the local clinical research ethics committee (MRC-05-161) and the Medical Research Center (MRC-01-20-679). The study passed through a fast track with approval number MRC-05-161, it was the initial approval from the hospital committee, and the IRB on 20 July 2020. It was only approved on an expedited basis; however, all the study documents were approved by the Medical Research Center (MRC) through the normal process with tracking number (MRC-01-20-679) on 24 July 2020. As per the approval, the study meets the waiver criteria of informed consent, and all data were fully anonymized before we accessed them.

### Outcomes

**1- Primary outcomes.**   Determine the mortality rate of dialysis patients with COVID-19 infection and associated risk factors. Mortality (death rate) was calculated per the following equation:

$$Death\ rate = \frac{Number\ of\ died\ dialysis\ patients\ with\ covid-19+ve}{Total\ Number\ of\ dialysis\ patients\ with\ covid-19+ve} \times 100$$

**2- Secondary outcomes.**   a-Determine the incidence of COVID-19 infection in dialysis patients in Qatar and assess its risk factors with comparison of COVID19 positive dialysis cohort to dialysis patients with no COVID19 (control).

b- Assess the severity of COVID-19 in dialysis patients and its related complications such as the incidence of hypoxia, ICU admission, need for mechanical ventilation, need for inotropes for resistant hypotension or shock, incidence of acute respiratory distress syndrome (ARDS), and length of hospital stay.

c- Compare the clinical outcomes of COVID-19 in peritoneal dialysis (PD) and HD patients.

### Inclusion and exclusion criteria

The study enrolled all eligible Dialysis patients in Qatar whether they undergo hemodialysis or peritoneal for dialysis treatment. To be eligible, the participant should be 18 years of age or older, diagnosed with end-stage renal disease that requires dialysis, receiving chronic maintenance hemodialysis or peritoneal dialysis for at least 1 month and diagnosed with acute COVID-19 infection. Participants were excluded if they are less than 18 years of age, had acute kidney injury that led hemodialysis requirement during COVID19 infection, or recent kidney transplant and stopped receiving dialysis, or an end stage renal disease started dialysis treatment for less than one month.

We only included those who a positive COVID-19 test from (February 2020 –July 2020).

## Study methodology and data collection

All data including demographics, clinical features, laboratory and radiological findings, treatment schemes, and mortality rates were collected from a national-based electronic medical record (Cerner-North Kansas City, MO, USA). Data has been accessed from 25 July- 25 October 2020. Methods for laboratory confirmation of COVID-19 infection have been described elsewhere [9]. The only method used in the state of Qatar by the Ministry of Public Health to diagnose COVID-19 is the gold standard, which is the polymerase chain reaction (PCR) assay of nasopharyngeal swab specimens following either routine screening (patients with exposure or at high risk) or acute presentation. Testing was performed at different locations (dialysis centers, emergency departments, and healthcare centers).

Routine blood examinations included complete blood count (CBC), coagulation profile (Prothrombin Time (PT), (partial thromboplastin time (PTT), International Normalized Ratio (INR), D-dimer (DD), and serum biochemistry including liver function tests, creatine kinase, LDH, total proteins, albumin, C-reactive protein (CRP), and ferritin. Laboratory parameters were taken upon diagnosis (baseline) and at 1 week after clinical onset. Peak values of different inflammatory markers such as white blood cells, peak serum ferritin level, peak CRP level, and peak interleukin-6 (IL-6) levels during COVID19 illness were also reported in this study.

We reviewed the chest X-ray reports of patients on admission and throughout their hospital stay. Chest radiography was classified as normal, unilateral pulmonary infiltrate, or bilateral pulmonary infiltrates.

## Dialysis scheme

During admission, all patients received 4-hour dialysis sessions, 3 times per week (our standard of care). The dialysis prescription was individualized according to the previous patient regimes and clinical status during admission.

## Statistical analysis

The qualitative variables are presented with their frequency distributions. Quantitative variables are summarized as mean ± SD or median and interquartile range. The association between qualitative variables was evaluated using Chi square or Fisher's exact test. Quantitative variables were analyzed using Student's t-test and/or an analysis of variance.

Univariate Cox regression analysis was used to explore the risk factors associated with in-hospital mortality. All statistical analyses were performed using SPSS software (version 21.0; Chicago, IL, USA). Statistical significance was defined as a 2-sided P value <0.05.

## Results

### 1- Patients characteristics, demographics and clinical presentation

Out of 1064 dialysis patients, 93/1064, (8.7%) were diagnosed with COVID-19; 76/93, (81.7%) of them (65/76, (85.5%) hemodialysis and 11/76 (14.5%) peritoneal dialysis) fulfilled the inclusion criteria of the study and were included in the analysis (Fig 1). In patients with COVID19 on dialysis, most were men (n = 56; 74%), most patients were from the Middle East (n = 34; 45%) or South Asian background (n = 26; 34%). The mean age of the patients was 56.5±13.6 years old. The most common comorbidities were hypertension (98.7%) and diabetes mellitus (65.7%). Most patients received influenza and pneumococcal vaccines (68% and 76%, respectively). The most common clinical presentations upon diagnosis were fever (57.9%), cough (56.6%), and shortness of breath (25%). The baseline characteristics of dialysis patients with COVID-19 are summarized in (Table 1).

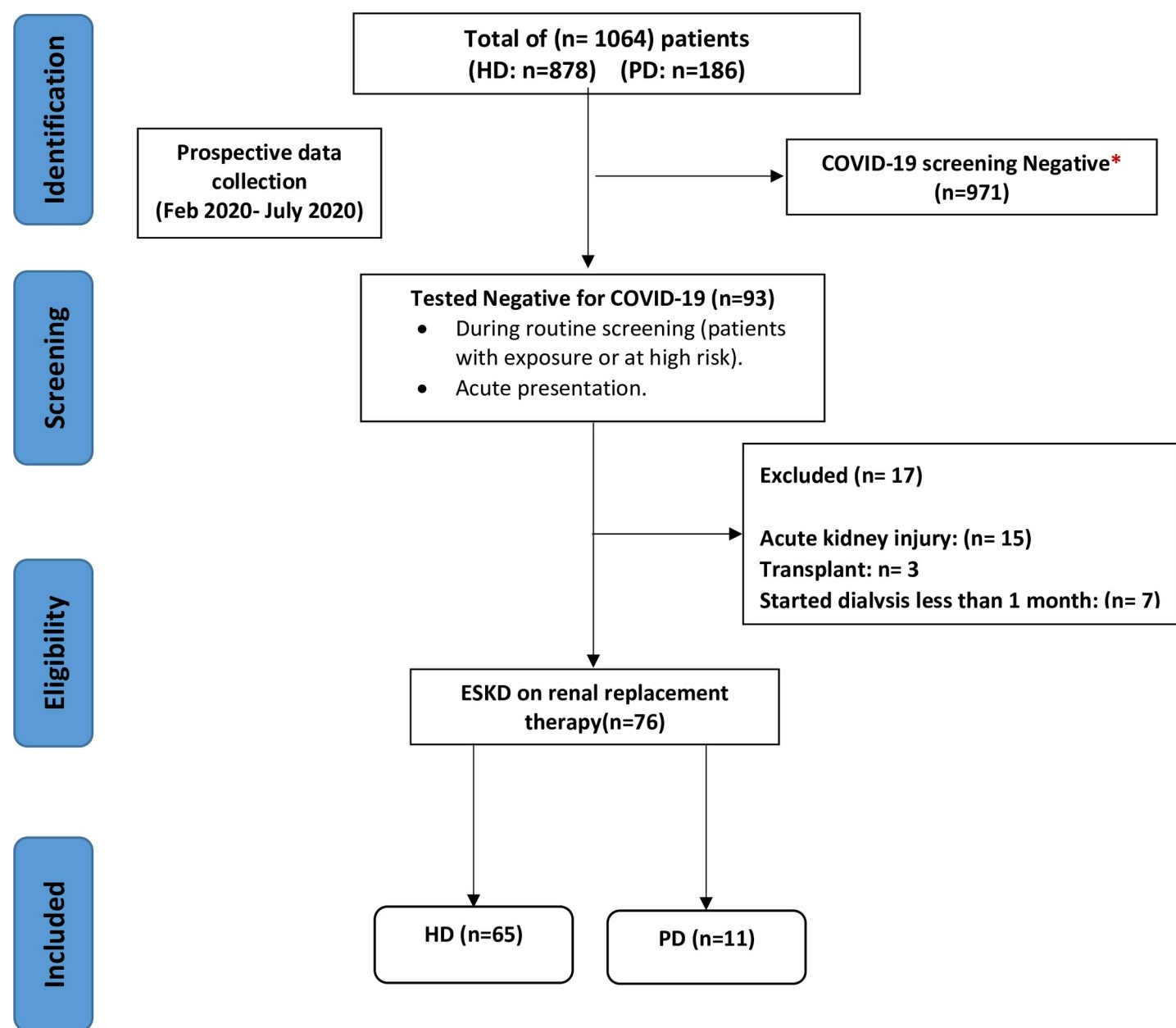

* COVID-19  test was negative or not tested ever.

HD, hemodialysis; PD, Peritoneal dialysis; HD/PD, patients receiving Hemodialysis and peritoneal dialysis.

**Fig 1. Study flow diagram shows total enrolled dialysis patients with COVID19 between February 1st, 2020 to July 19th, 2020.**

**Table 1. Shows demographics and comorbidities of dialysis patients with COVID-19 (study cohort) with comparison to dialysis patients without COVID19 (control group) in the state of Qatar.**

| | Dialysis with COVID19 (76 patients) | Dialysis without COVID19 (control) (988 patients) | P Value |
|---|---|---|---|
| **Age** | 56.5±13.6 years | 57.5+/-14.9 years | 0.432 |
| **Sex:** | | | |
| Male | 56 (73.6) | 567 (57.3) | |
| Female | 20 (26.3) | 421 (42.6) | 0.005 |
| **Ethnic group:** | | | |
| Middle east | 34 (44.7) | 731 (74) | 0.0001 |
| South Asia | 26 (34.2) | 148 (15) | 0.0001 |
| East Asia | 9 (11.8) | 89 (9) | 0.433 |
| Others | 7 (9.2) | 20 (2) | 0.0001 |
| **Source of SARS–Cov2 infection:** | | NA | |
| Recent travel | 2 (2.6) | | |
| Contact with patient | 18 (23.7) | | |
| Unknown | 56 (73.7) | | |
| **Comorbidities:** | | | |
| DM | 48 (65.7) | 632 (64) | 0.887 |
| Hypertension | 75 (98.7) | 968 (98) | 0.668 |
| IHD | 19 (25) | 257(26) | 0.846 |
| CHF | 2 (2.6) | 30 (3) | 0.842 |
| COPD | 2 (2.6) | 21 (2.1) | 0.770 |
| Asthma | 7 (9.2) | 27 (2.7) | 0.001 |
| H/O DVT | 4 (5.3) | * | |
| Atrial Fibrillation | 8 (10.5) | 30 (3) | 0.006 |
| **Dialysis modality:** | | | |
| HD | 65 (85.6) | 810 (82) | 0.436 |
| PD | 11 (14.4) | 178 (18) | |
| **SARS-Co2 symptoms at diagnosis:** | | NA | |
| Fever | 44 (57.9) | | |
| Cough | 43 (56.6) | | |
| GIT symptoms | 7 (9.2) | | |
| Sore throat | 8 (10.5) | | |
| SOB | 19 (25) | | |
| Myalgia | 1 (1.3) | | |
| Body pain | 4 (5.3) | | |
| Asymptomatic | 11 (14.5) | | |
| **Dialysis access"** | | | |
| CVC | 24 (31.5) | 275 (27.8) | 0.483 |
| AVF | 39 (51.3) | 512 (51.8) | 0.311 |
| AVG | 2 (2.6) | 23 (2.3) | 0.866 |
| PD | 11 (14.4) | 178 (18) | 0.436 |
| **H/o renal Transplantation** | 2 (5.3) | 46 (4.6) | 0.421 |
| H/O immunosuppression | 4 (2.6) | * | |
| Cyclosporine | 2 (2.6) | * | |
| Tacrolimus | 2 (2.6) | * | |
| Steroid | 2(2.6) | * | |
| H/O ACE/ARB pre-diagnosis | 11 (14.5) | * | |
| ACE/ARB held after diagnosis | 4 (5.3) | * | |

*(Continued)*

**Table 1.** (Continued)

| | Dialysis with COVID19 (76 patients) | Dialysis without COVID19 (control) (988 patients) | P Value |
|---|---|---|---|
| **Vaccination:** | | | |
| Flu vaccine | 52 (68.4) | 893 (92) | 0.0001 |
| Pneumococcal vaccine | 58 (76.3) | 951 (98) | 0.0001 |

NA data not applicable to this group.

* Data not available.

DM, diabetes mellitus; IHD, ischemic heart disease; CHF, congestive heart failure; COPD, Chronic obstructive pulmonary disease; DVT, deep vein thrombosis; HD, hemodialysis; PD, Peritoneal dialysis; SARS-Co2, Severe acute respiratory syndrome coronavirus 2 of the genus Betacorona virus; GIT, gastrointestinal tract, SOB, shortness of breath; CVC, central venous catheter; AVF, arteriovenous fistula; AVG, arteriovenous graft; ACE, angiotensin converting enzyme inhibitors; ARB, angiotensin-receptor blockers.

## 2- Primary outcomes

Eleven patients out of seventy-six (14.5%) died (non-survivor) of COVID-19 among our dialysis patients during study period. Non survivors dialysis patients with COVID-19 were significantly of older age (65.7-year-old in non-survivor group versus 54.9-year-old in survivor group; p = 0.005), had history of atrial fibrillation (36% versus 6%; p = 0.03), dialyzing using a central venous catheter (54% vs. 26%, p = 0.048), had a history of deep venous thrombosis (18% vs. 3%, p = 0.038), hypoxia upon presentation (82% versus 14%; p<0.001) and the presence of lung infiltrates on a chest x-ray (100% versus 68%; p = 0.05) compared to survivors. (Table 2). Non-survivors also had significantly lower day7 lymphocyte counts and higher values for ferritin, creatinine kinase (CPK), D-dimer, ALT, AST, and IL-6 levels and higher peak CRP, and higher baseline lactate and LDH in comparison to survivors. (Table 2).

Univariate Cox regression analysis (Table 3A) for risk factors associated with COVID-19-related death in dialysis patients showed statistically significant increases in risks with age (OR 1.077, CI 95%(1.018–1.139), p = 0.01), CHF and COPD (both same OR 8.974, CI 95% (1.039–77.5), p = 0.046), history of DVT (OR 5.762, CI 95% (1.227–27.057), p = 0.026), Atrial fibrillation (OR 7.285, CI 95%(2.029–26.150), p = 0.002), hypoxia (OR: 16.6; CI 95%(3.574–77.715), p = <0.001), ICU admission (HR30.8, CI 95% (3.9–241.2), p = 0.001), Mechanical ventilation (HR 50.07 CI 95% (6.4–391.2)), p<0.001) and using inotropes(HR 19.17, CI 95% (11.57–718.5), p<0.001). Laboratory values associated with significant risk for non-survivors were peak WBC peak level (OR 1.079, CI 95%(1.032–1.127), p = 0.001), fibrinogen level at day 7 (OR 1.28, CI 95%(1.065–1.545), p = 0.009) and D-dimer level both at baseline (OR 1.4, CI 95% (1.2–1.63) and on day 7 (OR 1.89, CI 95% (1.424–2.51).

The cox regression analysis summarized in Table 3A was a univariate analysis and did not account for potential cofounders. Therefore, we constructed in this revision a forward stepwise multivariate cox regression analysis model to further examine the association between the different statistically significant variables in the univariate analysis (Table 3A, p < 0.05) and mortality in dialysis patients infected with COVID-19 (Table 3B). The model initially included clinical variables that were statistically significant (p<0.05) and that were thought to be clinically relevant and not collinear. ICU admission was found to be the only clinical variable significantly associated with death [OR = 32.8 (3.5–305.4), p = 0.002)] (Table 3B). Then we tried to add the blood investigation variables that were statistically significant in the univariate analysis (Table 2A, p<0.05) to the model. However, the model became unstable due to our small sample size as well as the collinearity between ICU admission and inflammatory and thrombotic blood markers such as lactate, CRP, IL-6, D-dimer and ferritin.

**Table 2. Shows comparison of survivors to non-survivors with COVID19 using T-test for continuous variables and Chi-square statistical analysis for categorical variables.**

| Variable | Survivor (65) | Non survivor (11) | P |
|---|---|---|---|
| **Age** | 54.9±13.6 | 65.7±2.9 | 0.005 |
| **Gender:** | | | |
| Male | 46 | 10 | 0.161 |
| Female | 19 | 1 | |
| Diabetes | 43 (67.1) | 5 (45.4) | 0.188 |
| hypertension | 64 (98.4) | 11 (100) | 0.679 |
| **Nationality:** | | | |
| Citizens | 18 (27.6) | 2 (18.1) | 0.508 |
| Expatriates | 47 (72.3) | 9 (81.8) | |
| Ischemic Heart Disease | 14 (21.5) | 5 (45.4) | 0.09 |
| Congestive Heart Failure | 1 (1.5) | 1 (9.0) | 0.148 |
| COPD | 1 (1.5) | 1 (8.0) | 0.148 |
| Atrial Fibrillation | 4 (6.15) | 4 (36.3) | 0.03 |
| Bronchial Asthma | 5 (7.69) | 2 (18.1) | 0.266 |
| History of deep venous thrombosis | 2 (3.0) | 2 (18.1) | 0.038 |
| Received Flu Vaccine | 42(64.6) | 10 (90.9) | 0.083 |
| Received pneumococcal vaccine | 51 (78.4) | 7 (63.6) | 0.285 |
| **Dialysis Modality:** | | | |
| Hemodialysis | 55 (84.6) | 10 (90.9) | 0.5 |
| Peritoneal dialysis | 10 (65.0) | 1 (9.0) | |
| **Dialysis Access:** | | | |
| Central venous catheter | 18 (27.7) | 6 (54.5) | 0.048 |
| Arteriovenous fistula | 35 (53.8) | 4 (36.3) | |
| Arteriovenous graft | 2 (3.0) | 0 (0) | |
| Peritoneal dialysis | 10 (15.3) | 1(9) | |
| History of kidney transplant | 3 (4.6) | 1 (9.0) | 0.539 |
| Steroid | 1 (1.5) | 1 (9.0) | 0.148 |
| **Symptoms at diagnosis:** | | | |
| SOB | 1 | 1 | 0.195 |
| Fever | 38 | 6 | 0.970 |
| Cough | 34 | 9 | 0.028 |
| Sore throat | 7 | 1 | 0.350 |
| Myalgia | 1 | 0 | 0.326 |
| Body pain | 4 | 0 | 0.257 |
| Vomiting | 6 | 1 | 0.350 |
| Diarrhea | 6 | 1 | 0.950 |
| Hypoxia | 9 (13.8) | 9 (81.8) | <0.001 |
| Admission to ICU | 10(15) | 10(90) | <0.0001 |
| Mechanical ventilation | 4(6) | 10(90) | <0.0001 |
| Inotropes | 1(2) | 10(90) | <0.0001 |
| **Chest x ray:** | | | 0.05 |
| Normal | 21 (32.3) | 0 (0) | |
| Unilateral | 9 (13.8) | 1 (9.0) | |
| Bilateral | 35 (53.8) | 10 (90.9) | |
| WBC base | 6.7±2.8 | 6.0±3.1 | 0.598 |
| WBC after 7 days | 6.2±4.1 | 9±4.5 | 0.269 |

(*Continued*)

**Table 2.** (Continued)

| Variable | Survivor (65) | Non survivor (11) | P |
|---|---|---|---|
| WBC peak | 9.9±7.6 | 21.9±10 | 0.058 |
| Lymphocytes base | 1.2±0.63 | 0.88±0.65 | 0.945 |
| Lymphocytes after 7 days | 1.07±0.76 | 0.58±0.25 | 0.019 |
| HB base | 11.5±1.6 | 11.3±1.7 | 0.583 |
| HB after 7 days | 10.1±3.2 | 10.7±2.0 | 0.656 |
| PLT base | 221.1±262 | 164±86.2 | 0.787 |
| PLT after 7 days | 207.8±103.9 | 149.9±96.7 | 0.497 |
| PT base | 6.9±7.2 | 18.2±11.9 | 0.218 |
| PT after 7 days | 7.3±13.8 | 20.4±11.8 | 0.616 |
| PTT base | 19.3±19.0 | 38.6±10.0 | <0.001 |
| PTT after 7 days | 13.7±17.2 | 47.6±25.3 | 0.466 |
| INR base | 0.59±0.61 | 1.5±0.99 | 0.172 |
| INR after 7 days | 0.54±1.0 | 1.7±0.99 | 0.388 |
| Ferritin base | 1188.4±230.1 | 4774.4±9357.3 | <0.001 |
| Ferritin after 7 days | 1492.5±2865.1 | 11923.5±20982 | <0.001 |
| Ferritin Peak | 2469.4±4792.7 | 23575±22875.1 | <0.001 |
| Fibrinogen base | 1.83±6.0 | 3.2±2.2 | 0.597 |
| Fibrinogen after 7 days | 1.3±2.4 | 3.8±2.2 | 0.422 |
| CRP base | 41.6±63.6 | 77.2±83.4 | 0.192 |
| CRP after 7 days | 45.6±68.3 | 113.4±85.9 | 0.214 |
| CRP peak | 74.5±94.5 | 477.1±871.2 | <0.001 |
| CPK base | 126.1±516.4 | 883.1±1699.9 | <0.001 |
| CPK after 7 days | 15.4±43.1 | 632±1849.4 | <0.001 |
| LDH base | 136.5±169.8 | 424.1±365.3 | 0.025 |
| LDH after 7 days | 106.5±194.2 | 374.2±286.3 | 0.054 |
| D-Dimer base | 0.797±1.23 | 12.0±18.8 | <0.001 |
| D-Dimer after 7 days | 0.569±0.90 | 3.0±2.2 | <0.001 |
| Lactate base | 0.42±0.68 | 1.2±1.8 | <0.001 |
| Lactate after 7 days | 1.5±4.9 | 0.38±0.67 | 0.086 |
| ALT base | 20.6±15.0 | 40.1±50.9 | <0.001 |
| ALT after 7 days | 20.7±14.2 | 251±709 | <0.001 |
| AST base | 24.7±15.7 | 77.3±83.5 | <0.001 |
| AST after 7 days | 25.1±18.4 | 696.2±2091 | <0.001 |
| IL-6 Base | 16.7±40.9 | 340.2±918.7 | 0.001 |
| IL-6 after 7 days | 57.6±350.7 | 66.8±132.9 | 0.952 |
| IL-6 Peak | 460.7±2056 | 3270±5240 | <0.001 |
| O2 saturation base | 95.7±8.1 | 93.6±18.4 | 0.533 |
| O2 saturation after 7 days | 94.0±8.4 | 93.4±4.3 | 0.977 |
| Albumin base | 32.6±6.7 | 24.4±11.4 | 0.091 |
| Albumin after 7 days | 27.6±16.1 | 23.5±5.5 | 0.790 |

## 3- Secondary outcomes

**A- Incidence.** Seventy-six patients out of 1064 total dialysis patients (7.1%) were diagnosed with COVID-19 in the study period. (Fig 2) shows accumulative cases of COVID-19 (total) and monthly incidence of COVID-19 positive dialysis patients in the state of Qatar during the study period.

**Table 3. Shows hazard ratios for risk factors in non-survivor group using (A) univariate and (B) multivariate.**
Cox regression analysis. A. shows hazard ratios for risk factors in non-survivor group using univariate Cox regression analysis. B. A forward stepwise multivariate cox regression analysis model to examine the association between the different statistically significant variables in the univariate analysis (Table 3A, p < 0.05) and mortality in dialysis patients infected with COVID-19.

| Variable | HR | 95% CI | P |
|---|---|---|---|
| **A** | | | |
| Age | 1.077 | 1.018–1.139 | 0.010 |
| Gender | 3.112 | 0.389–24.3 | 0.279 |
| Ethnic group | 0.646 | 0.296–1.1412 | 0.273 |
| Diabetes | 0.503 | 0.153–1.651 | 0.257 |
| Hypertension | 21.081 | 0.142–84.742 | 0.733 |
| IHD | 2.485 | 0.750–8.109 | 0.137 |
| CHF | 8.974 | 1.039–77.508 | 0.046 |
| COPD | 8.974 | 1.039–77.508 | 0.046 |
| H/o DVT | 5.762 | 1.227–27.057 | 0.026 |
| AF | 7.285 | 2.029–26.150 | 0.002 |
| Flu vaccine | 5.336 | 0.682–41.720 | 0.111 |
| Dialysis modality | 1.250 | 0.158–9.895 | 0.833 |
| Dialysis Access | 0.719 | 0.337–1.536 | 0.395 |
| ACEi use | 1.517 | 0.356–7.051 | 0.595 |
| Hypoxia | 16.666 | 3.574–77.715 | <0.001 |
| X RAY findings | 4.234 | 0.820–21853 | 0.085 |
| ICU admission | 30.823 | 3.93–241.211 | 0.001 |
| Mechanical ventilation | 50.074 | 6.408–391.294 | <0.001 |
| Using Inotropes | 19.178 | 11.57–718.503 | <0.001 |
| WBC | 0.887 | 0.702–1.120 | 0.312 |
| WBC peak | 1.079 | 1.032–1.127 | 0.001 |
| Lymphocytes | 0.341 | 0.008–1.318 | 0.119 |
| Lymphocytes peak | 0.371 | 0.130–1.071 | 0.067 |
| HB | 0.973 | 0.675–1.402 | 0.882 |
| Platelets | 0.997 | 0.989–1.005 | 0.465 |
| Ferritin | 1.000 | 1.000–1.000 | 0.053 |
| Ferritin peak | 1.000 | 1.000–1.000 | <0.001 |
| Fibrinogen | 1.019 | 0.957–1.085 | 0.561 |
| Fibrinogen 7 days | 1.283 | 1.065–1.545 | 0.009 |
| CRP base | 1.005 | 0.999–1.011 | 0.114 |
| CRP peak | 1.001 | 1.000–1.001 | 0.020 |
| CK base | 1.001 | 1.000–1.001 | 0.001 |
| CK after 7 days | 1.010 | 1.004–1.017 | 0.001 |
| LDH base | 1.002 | 1.001–1.004 | <0.001 |
| LDH after 7 days | 1.003 | 1.001–1.006 | 0.002 |
| D-Dimer base | 1.404 | 1.209–1.631 | <0.001 |
| D-Dimer 7 days | 1.891 | 1.424–2.512 | <0.001 |
| IL-6 base | 1.001 | 1.000–1.001 | 0.037 |
| IL-6 peak | 1.000 | 1.000–1.000 | 0.013 |
| Albumin base | 0.962 | 0.916–1.010 | 0.117 |
| Albumin 7 days | 1.004 | 0.955–1.057 | 0.866 |
| ALT base | 1.014 | 1.002–1.026 | 0.026 |
| ALT after 7 days | 1.002 | 1.001–1.003 | 0.002 |

(*Continued*)

**Table 3.** (Continued)

| Variable | HR | 95% CI | P |
|---|---|---|---|
| AST base | 1.014 | 1.006–1.021 | >0.001 |
| AST after 7 days | 1.001 | 1.000–1.000 | 0.002 |
| Lactate base | 1.867 | 1.269–2.748 | 0.002 |
| Lactate 7 days | 0.897 | 0.674–1.094 | 0.455 |
| **B** | | | |
| Age | 1.02 | (0.95, 1.09) | 0.62 |
| H/o DVT | 1.93 | (0.14, 25.87) | 0.62 |
| AF | 2.15 | (0.26, 17.67) | 0.48 |
| ICU Admission | 32.75 | (3.51, 305.35) | 0.002 |

COPD, Chronic obstructive pulmonary disease; SOB, shortness of breath; WBC, white blood cells; HB, hemoglobin; PLT, platelets; PT, prothrombin time; PTT, partial thromboplastin time; INR, international normalized ratio; CRP, c-reactive protein; CPK, Creatine phosphokinase; CK, creatine kinase; LDH, Lactate dehydrogenase; ALT, alanine transaminase; AST, aspartate aminotransferase; IL, interleukin; O2, oxygen.

Compared to patients on dialysis in Qatar who did not have COVID-19 (control group), our cohort with COVID-19 had significantly more males, South Asian background, asthma and atrial fibrillation, and less patients of Middle East background and vaccination for influenza or pneumonia. Most comorbid conditions, age and vascular access type where not different between the two groups. Significantly more men than women on dialysis had COVID-19 (9% of all male dialysis population (56 out of 623) versus 4.5% in females (20 out of 441; p = 0.015, OR 1.89, p = 0.017). Table 1 details comparison of our COVID-19 positive dialysis patients versus COVID negative dialysis patients (control).

**B- Severity and complications.** Although only 18 (23.7%) of COVID-19 patients had documented hypoxia (O2 saturation ≤ 95%), 55(72.4%) had pneumonia during their COVID-19 course (mostly bilateral (45(82%) of them). Only 21 patients (27.6%) had normal chest X

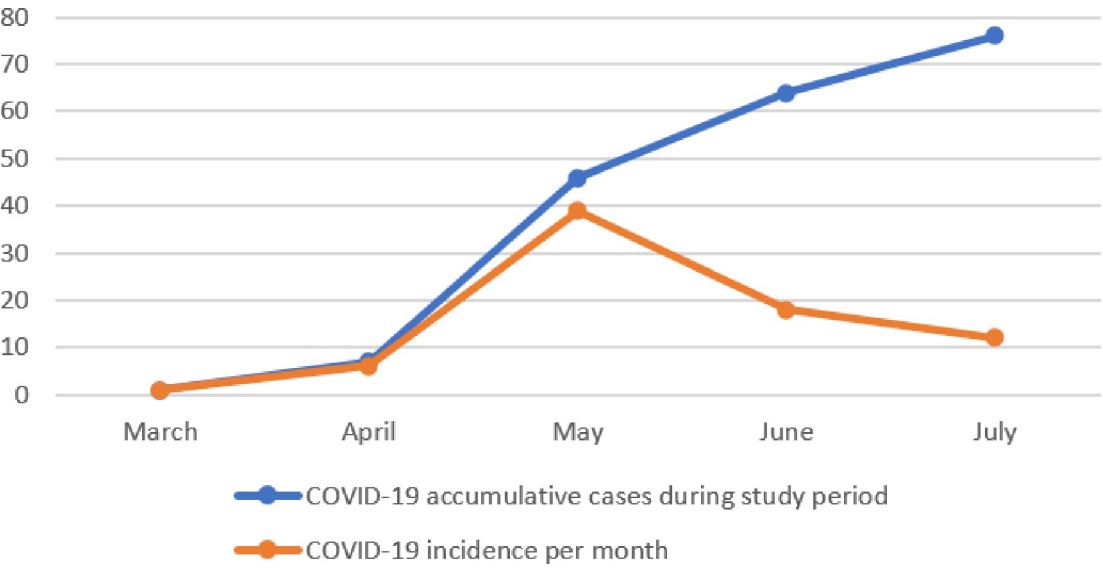

**Fig 2. Accumulative cases and monthly incidence of COVID-19 positive dialysis patients in the state of Qatar during study period.**

ray. Nineteen patients (25%) required admission to the intensive care unit (with length of stay (LOS) in ICU of 15+/-11 days before death or transfer to general floor); 14 patients (18.4%) had ARDS, 13 (17.1%) required mechanical ventilation, 11 patients (14.5%) required inotropes to treat severe hypotension and 1 patient (1.3%) had deep venous thrombosis (DVT). Length of hospital stay varied with a mean of 19.2 ± -12 days. Patients admitted to ICU had longer total hospital LOS compared to patients admitted to general floor (31.5+/-8 versus 17.1+/-11 respectively, p = 0.0014). Laboratory tests showed some trend for increase in inflammatory markers in day 7 and peak values compared to baseline values. Fig 3 shows baseline laboratory test that changed significantly compared to either day7 (lymphocyte and Albumin levels) or peak value (WBC, ferritin, CRP and IL-6).

**C- Comparison of PD versus HD.** Eleven PD patients versus 65 HD patients were diagnosed with COVID-19. COVID19 infection affected 5.9% (11/186) of all patients with PD versus 7.4% (65/878) of the total HD patients (p = 0.3). See (Tables 1 and 4) and (Fig 1). Mortality among COVID-19 positive cases was 9%(1/11) in PD compared to 15%(10/76) in HD (p = 0.5). Table 4 showed not statistically significant different among PD and HD COVID-19 patients in most characteristics (demographics, comorbidities, presenting symptoms, primary and secondary outcomes etc.) except for more diarrhea in PD patients (p = 0.029). Interestingly, PD COVID-19 patients had statistically significant higher peak WBC counts and higher creatinine kinase, ferritin, LDH, lactate, ALT, AST, and IL-6 levels upon admission (base) compared to HD COVID-19 patients.

## Discussion

In this study, we described the clinical course and outcome of COVID-19 infection in dialysis patients in Qatar. To the best of our knowledge, this is the first study to be conducted at a national level considering the effect of COVID-19 on the dialysis population in an entire country.

The main risk groups for mortality and developing complications during the COVID-19 pandemic are the elderly and people with chronic health conditions [9]. Dialysis patients are expected to be more likely to develop COVID-19, given their limited ability to self-isolate as they require frequent visits to health care facilities. Our dialysis patients had almost twice the number of COVID-19 patients compared to the general population in Qatar. Incidence of COVID19 was 7.1% in our dialysis cohort versus 4% in nationwide [10]. They are also expected to suffer more complications and mortality, given their age and comorbidities. Our dialysis patients had approximately a hundred times higher risk of death compared to the general population in Qatar based on available national data (15% in our dialysis cohort versus 0.15% countrywide) [10].

The mean age of dialysis patients with COVID-19 was 56.5±13.6 years compared to the mean age of 58 years in the dialysis population in Qatar [11]. Approximately three-quarters of the infected patients were males, although they represent just over half of our dialysis cohort. This is probably because males in Qatar are more socially active, as they are breadwinners. Women usually abide to the COVID-19 preventative measures [12]. In our study, males tended to have a higher risk of death, which replicates data from previous reports [13]. In the deceased patient group, there was a trend for more males than females, but the difference was not statistically significant ($P = 0.161$).

Patients from the Middle East had a lower incidence of COVID-19 infection compared to patients from South Asia or East Asia in our study. This is likely to be explained by the effect of socioeconomic status. Oh et al. found that lower socioeconomic status was associated with a higher risk of contracting COVID-19 in South Korea especially in the older population [14]

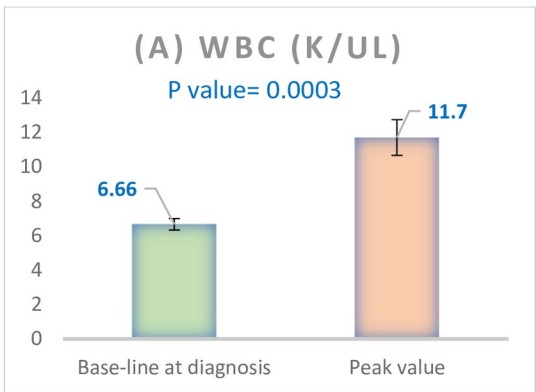

**A**: Mean WBC (± SE) Comparison of baseline and peak value.

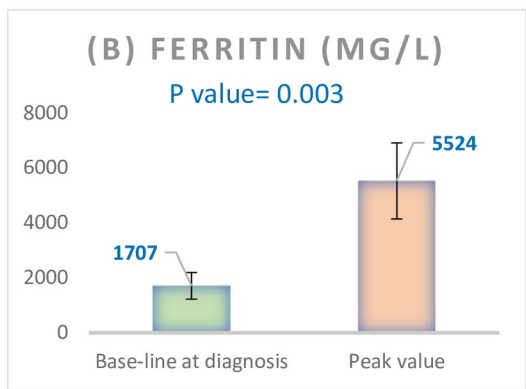

**B**: Mean Ferritin (± SE) Comparison of baseline and peak value.

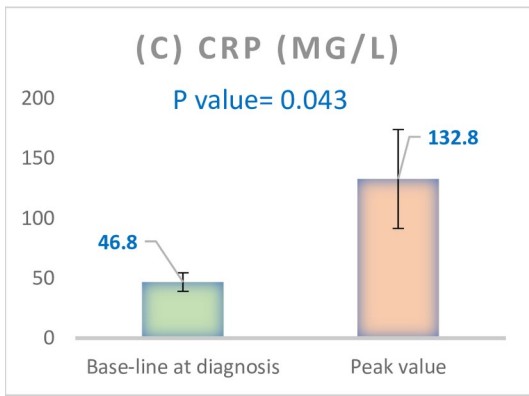

**C**: Mean CRP (± SE) Comparison of baseline and peak value.

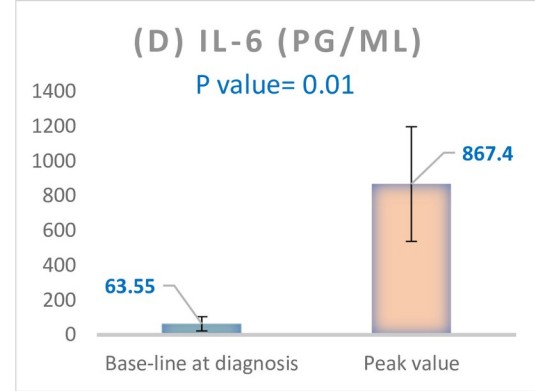

**D**: Mean IL-6 (± SE) Comparison of baseline and peak value.

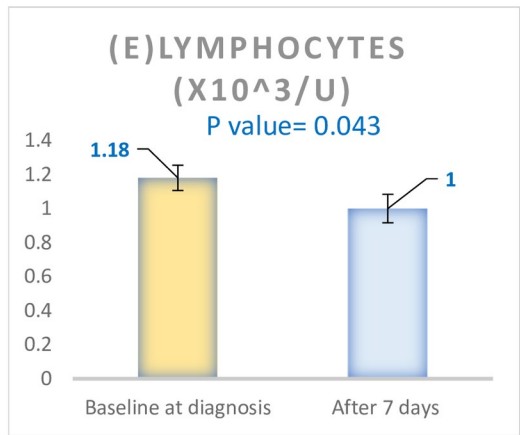

**E**: Mean Lymphocytes (± SE) Comparison of baseline and after 7 days value.

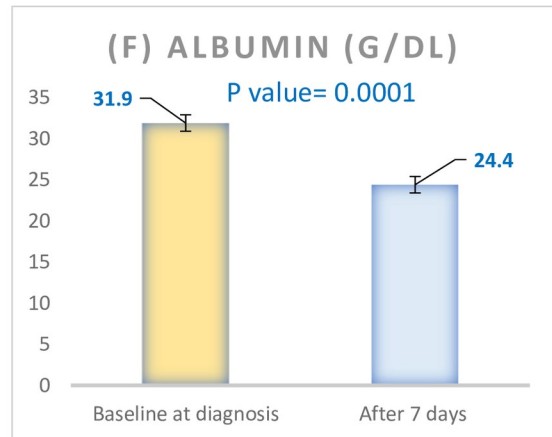

**F**: Mean Albumin (± SE) Comparison of baseline and after 7days value.

**Fig 3. Laboratory tests of dialysis patients with COVID19 with comparison of baseline and day 7 (for lymphocyte and Albumin levels) and baseline and peak value (for WBC, ferritin, CRP and IL-6).**

**Table 4. Demographic, clinical, laboratory and radiological features in hemodialysis (HD) and peritoneal dialysis (PD) patients.**

| | HD (65) | PD (11) | P |
|---|---|---|---|
| Age | 65.6±14.3 | 55.4±8.5 | 0.108 |
| Male/female | 47/18 | 9/2 | 0.508 |
| **Ethnic groups:** | | | |
| Middle east | 33 (50.7) | 1 (9.0) | 0.014 |
| South Asia | 21 (32.3) | 5 (45.4) | |
| East Asia | 5 (7.6) | 4 (36.3) | |
| Others | 6 (9.2) | 1 (9.0) | |
| **Covid19 source:** | | | |
| Recent travel | 2 (3.0) | 0 | 0.734 |
| Close contact | 16 (24.6) | 2 18.1) | |
| Unknown | 47 (72.3) | 9 (81.8) | |
| **Comorbidities:** | | | |
| Diabetes | 40 (61.5) | 8 (72.7) | 0.477 |
| Hypertension | 64 (98.4) | 11 (100) | 0.679 |
| IHD | 17 (26.1) | 2 (18.1) | 0.572 |
| CHF | 2 (3.0) | 0 | 0.555 |
| COPD | 2 (3.0) | 0 | 0.555 |
| Asthma | 7 (10.7) | 0 | 0.253 |
| H/o DVT | 4 (6.1) | 0 | 0.398 |
| AF | 8 12.3) | 0 | 0.219 |
| H/o Flu vaccine | 43 (66.1) | 9 (81.8) | 0.301 |
| H/o Pneumonia vaccine | 46 (70.7) | 10 (90.9) | 0.218 |
| H/o Renal Tx | 4 (6.1) | 0 | 0.398 |
| ACEI before Covid | 3 (4.6) | 8 (72.7) | <0.001 |
| ACEI held after infection | 0 | 4 (36.3) | <0.001 |
| **Covid19 Symptoms:** | | | |
| SOB | 2 (3.0) | 0 | 0.566 |
| Fever | 38 (58.4) | 6 (54.5) | 0.719 |
| Cough | 36 (55.3) | 7 (63.6) | 0.687 |
| Vomiting | 2 (3.0) | 0 | 0.467 |
| Diarrhea | 4 (6.1) | 3 (27.2) | 0.029 |
| Sore throat | 7 (10.7) | 1 (9.0) | 0.824 |
| Body pain | 4 (6.1) | 0 | 0.576 |
| Myalgia | 0 | 1 (9.0) | 0.043 |
| Asymptomatic | 11 (16.9) | 0 | 0.133 |
| Hypoxia | 14 (21.5) | 4 (36.3) | 0.285 |
| **Outcomes:** | | | |
| Mortality | 10 (15.3) | 1 (9.0) | 0.583 |
| ICU admission | 16 (24.6) | 3 (27.2) | 0.833 |
| ARDS | 1 (1.5) | 0 | 0.668 |
| DIC | 0 | 1 (9.0) | 0.014 |
| Ventilation | 12 (18.4) | 1 (9.0) | 0.674 |
| Inotropes | 10 (15.3) | 1 (9.0) | 0.583 |
| **Chest X ray:** | | | |
| No changes | 18 | 3 (27.2) | 0.903 |
| Unilateral infiltrate | 9 | 1 (9.0) | |
| Bilateral infiltrate | 38 | 11 (100) | |

*(Continued)*

**Table 4.** (Continued)

| | HD (65) | PD (11) | P |
|---|---|---|---|
| Hospital stay (days) | 19.0±13.1 | 19.5±8.6 | 0.134 |
| Quarantine days | 5.4±9.3 | 1.8±4.8 | 0.043 |
| O2 Saturation base % | 95.9±18.4 | 93.3±2.2 | 0.320 |
| O2 saturation 7 days % | 92.9±18.4 | 97.3±2.2 | 0.212 |
| WBC base | 6.5±2.7 | 7.5±7.7 | 0.181 |
| WBC after 7 days | 6.0±3.6 | 10.1±6.2 | 0.012 |
| WBC peak | 11.0±8.1 | 15.7±12.8 | 0.014 |
| Lymphocyte base | 1.17±0.659 | 1.2±0.603 | 0.754 |
| Lymphocyte after 7 days | 1.02±0.751 | 0.89±0.61 | 0..498 |
| HB base | 11.5±1.7 | 11.4±1.03 | 0.170 |
| HB after 7 days | 10.1±3.3 | 10.5±1.0 | 0.081 |
| Platelets base | 213.1±263.9 | 211.1±64.5 | 0..564 |
| Platelets after 7 days | 193.6±102.3 | 233.9±114.4 | 0.657 |
| PT base | 8.7±9.3 | 7.7±6.1 | 0.363 |
| PT after 7 days | 9.6±15.2 | 6.7±6.5 | 0.352 |
| PTT base | 22.2±19.6 | 21.4±17.0 | 0.384 |
| PTT after 7 days | 18.4±21.9 | 19.6±23.2 | 0.817 |
| INR base | 0.74±0.79 | 0.65±0.52 | 0.330 |
| INR after 7 days | 0.73±1.15 | 0.56±0.54 | 0.378 |
| Ferritin base | 1424.1±2484.3 | 3381.7±9510.9 | 0.001 |
| Ferritin after 7 days | 3283.8±9581.3 | 1338.0±2021.0 | 0.301 |
| Ferritin peak | 5665.3±12153.5 | 4690.0±11963.3 | 0.895 |
| Fibrinogen base | 1.9±6.0 | 2.3±2.7 | 0.949 |
| Fibrinogen after 7 days | 1.6±2.4 | 2.4±3.1 | 0.191 |
| CRP base | 47.3±70.2 | 44.3±49.2 | 0.634 |
| CRP 7 after 7 days | 57.9±78.7 | 40.7±41.3 | 0.140 |
| CRP peak | 138.8±387.3 | 97.7±83.4 | 0.484 |
| CK base | 151.9±689.2 | 730.9±1334.2 | 0.002 |
| CK after 7 days | 111.3±768.1 | 652.0±107.3 | 0.632 |
| LDH base | 148.7±170 | 350.6±415.3 | 0.006 |
| LDH after 7 days | 131.8±216.5 | 224.9±287.7 | 0.102 |
| D-Dimer base | 2.6±8.6 | 1.1±2.0 | 0.361 |
| D-Dimer after 7 days | 0.88±1.4 | 1.14±1.6 | 0.546 |
| Lactate base | 0.48±0.87 | 1.0±1.4 | 0.016 |
| Lactate after 7 days | 1.59±4.9 | 0.41±0.72 | 0.094 |
| ALT base | 21.2±16.9 | 36.8±48.2 | 0.003 |
| ALT after 7 days | 55.6±293.8 | 44.5±52.8 | 0.649 |
| AST base | 29.9±34.9 | 47.0±55.3 | 0.048 |
| AST after 7 days | 135.6±865.0 | 43.7±31.0 | 0.459 |
| IL-6 base | 20.9±57.8 | 315.5±922.0 | <0.001 |
| IL6 after 7 days | 64.2±354.2 | 27.8±30.6 | 0.339 |
| IL-6 peak | 759.5±2443.2 | 1504.8±4820.6 | 0.091 |
| Albumin base | 32.6±9.1 | 28.0±4.4 | 0.216 |
| Albumin after 7 days | 24.3±11.5 | 24.6±6.5 | 0.031 |

IHD, ischemic heart disease; CHF, congestive heart failure; COPD, Chronic obstructive pulmonary disease; DVT, deep vein thrombosis; AF, atrial fibrillation; Angiotensin-converting enzyme inhibitor (ACEI) s; WBC, white blood cells; HB, hemoglobin; PLT, platelets; PT, prothrombin time; PTT, partial thromboplastin time; INR, international normalized ratio; CRP, c-reactive protein; CPK, Creatine phosphokinase; CK, creatine kinase; LDH, Lactate dehydrogenase; ALT, alanine transaminase; AST, aspartate aminotransferase; IL, interleukin; O2, oxygen.

while Hawkins et al. found that lower education levels, median income and poverty rate were strongly associated with higher rate COVID-19 cases [15].

Comorbidity profiles and dialysis modality were comparable in our general dialysis population [16]. PD patients had trend toward less incidence and lower mortality rates than HD patients but was not statistically significant. This result though is consistent with the study by M Sachdeva et al. which suggested that hospitalized patients on PD had a relatively mild course [17]. The only PD patient who died had PD-related fungal infection, and his PD catheter was removed before switching to HD.

About two-thirds of the patients who were on renin-angiotensin-aldosterone system inhibitors continued on this class of medications. This is in keeping with recent evidence suggesting that renin-angiotensin-aldosterone-system inhibitors are associated with reduced mortality in patients with sepsis [18].

Our patients presented with a similar profile of clinical symptoms compared to other diagnosed patients in the country [19]. Three-quarters of patients had a radiological evidence of the disease in their lungs. Most patients experienced bilateral changes. This is similar to previous report by Vancheri et al. [20] who described that among 240 patients with COVID19 who underwent chest X ray, 73.3% showed bilateral lung alteration (infiltrates, reticular or ground glass opacity) with only 25% showed negative X ray.

PD patients had similar demographics, comorbidities, presenting symptoms (except for more diarrhea), and secondary outcomes including hospital length of stay compared to HD patients. We noticed that our PD COVID-19 patients had statistically significant higher peak WBC counts and higher base creatinine kinase, ferritin, LDH, lactate, ALT, AST, and IL-6 levels compared to HD COVID-19 patients. This unexpected observation did not reflect differences in outcomes between the two groups and need to be studied on a larger group to confirm this finding.

Among all comorbidities, only atrial fibrillation and deep venous thrombosis were significantly associated with mortality. The percentage of AF in non-survivors was higher than that reported previously in Italy [21]. This could be due to the small sample size. As expected, non-survivors had evidence of association with multiple inflammatory markers (low lymphocyte count and high levels of ferritin, peak CRP, Creatine Kinase, IL6, D-dimer, AST and ALT). Mortality was associated with older age and dialysis via a central venous catheter in hemodialysis patients. Our patients with central venous catheters tended to be generally older and frailer. Although Qatar has one of the lowest COVID-19-related mortality rates in the world (0.15%), the mortality rate among our dialysis patients was significantly higher (14.5%). Still, this remains lower than most international figures reported in Spain, Italy, and Turkey (Mortality rate of 30.4%, 40% and 13% respectively) [22–24].

Our study has some limitations, the most important of which is its observational nature. Some clinical data, such as symptoms at presentation, may have been missing. It is also likely that we missed asymptomatic patients who might constitute up to 25% according to a systematic review by W Koh et al. [25]. There was no screening program that might have included these patients. Despite this, it is the only study on COVID-19 and dialysis at a national level. It is also unique in that it includes dialysis patients with both modalities.

## Conclusion

We are presenting the first study on COVID-19-related outcomes in dialysis patients in the state of Qatar. There was a high incidence of COVID-19 infection and a much higher mortality rate compared to the general population of Qatar. Dialysis patients also had prolonged hospitalization and multiple COVID-19-related complications. Special care is needed to prevent COVID-19 infection in dialysis patients to prevent this severe course and outcomes.

## Supporting information

**S1 Data.**
(XLSX)

## Acknowledgments

We profusely thank all the contributors from HMC for their excellent efforts and continue to support Ms. Mathew M (RN), Mr. Aly S (RN), Dr. Tawhid H (MD), Ms. YASIN S (RN), Ms LONAPPAN V (RN), Mr Farooqi F (CH) Ateya H (RN). We greatly thank Dr. Mohamed Elshazly for performing further statistical analysis. We would like to thank Editage (www. editage.com) for English language editing.

## Author Contributions

**Conceptualization:** Tarek Abdel Latif Ghonimi, Mohamad Mahmood Alkad, Essa Abdulla Abuhelaiqa, Muftah M. Othman, Musab Ahmed Elgaali, Abdullah Ibrahim Hamad.

**Data curation:** Tarek Abdel Latif Ghonimi, Rania Abdelaziz M. Ibrahim, Abdullah Ibrahim Hamad.

**Formal analysis:** Tarek Abdel Latif Ghonimi.

**Investigation:** Mohamad Mahmood Alkad, Muftah M. Othman, Rania Abdelaziz M. Ibrahim, Shajahan M. Joseph, Abdullah Ibrahim Hamad.

**Methodology:** Tarek Abdel Latif Ghonimi, Mohamad Mahmood Alkad, Muftah M. Othman, Abdullah Ibrahim Hamad.

**Resources:** Muftah M. Othman.

**Supervision:** Hassan Ali Al-Malki.

**Writing – original draft:** Abdullah Ibrahim Hamad.

**Writing – review & editing:** Mohamad Mahmood Alkad, Essa Abdulla Abuhelaiqa, Rania Abdelaziz M. Ibrahim, Hassan Ali Al-Malki.

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
