## [Decision Letter · Decision Letter 0]

17 Mar 2021

PONE-D-21-00180

Incidence and Outcomes of COVID19 Infection in Dialysis Patients in Qatar: A nationwide Cohort Study.

PLOS ONE

Dear Dr. Hamad,

Thank you for submitting your manuscript to PLOS ONE. After careful consideration, we feel that it has merit but does not fully meet PLOS ONE’s publication criteria as it currently stands. Therefore, we invite you to submit a revised version of the manuscript that addresses the points raised during the review process.

We look forward to receiving your revised manuscript.

Kind regards,

Bijan Najafi

Academic Editor

PLOS ONE

Additional Editor Comments:

Thanks for contributing your original study to PLOS One. It was reviewed by two experts with complementary expertise relevant to the scope of this study. Both reviewers agreed that the study is novel, has potential impact and high significance. There were however few concerns voiced by both reviewers which should be addressed before recommending the manuscript for acceptance in PLOS One. Please note that some of the specific comments of the reviewer #2 were placed by mistake in another section that may not be visible to you. I requested the editorial office to make those comments visible to author. In addition, I listed those comments in the following to avoid further delay in decision:

"Additional Comments from the Reviewer#2"

The authors explain the incidence and mortality of dialysis patients with COVID-19 as well as associated risk factors

The main claims of the paper are predictors of mortality of covid-19 infection in patients with dialysis in the state of Qatar. However, regression analysis did not include potential confounders that may have caused influence with associated mortality. The baseline characteristics significantly different between groups should undergo a selection method (i.e., pairwise, forward, etc) to be included as covariates for regression analysis.

Technical clarifications: The authors should clarify the definition of general population. It seems that this term was used for the control group as means of dialysis patients without covid-19 infection. There are sections of the text where the authors mention general population as the overall population in Qatar, regardless of dialysis. Incidence should be calculated between those patients under dialysis with and without covid-19, and mortality between survivors vs non-survivors under dialysis infected with covid-19.

Data presentation: This study requires more clarity on weather mortality association was assessed with chi2 or univariate regression analysis.

1. TITLE

Please include mortality in the title since the paper is focused in associated death risk.

2. ABSTRACT

Methods: Recommended to emphasize these patients were on maintenance dialysis therapy. Please mention univariate regression analysis in this section since line 97 is a result of this test.

Results: Line 89: Do the authors refer as patients under dialysis without covid-19 as general population?

3. INTRODUCTION

Line 129: Does general population mean patients under dialysis? The abstract is using “general population” as control, thus, it is understood that those patients were also under dialysis.

Line 141: Recommended to not include objectives in this section since this overlap with the primary and secondary outcomes mentioned in the methods section. Recommended to substitute with “The purpose of this study is to determine the incidence and mortality of covid-19 infection in patients under dialysis in the state of Qatar".

4. METHODS

Please describe the exclusion criteria from Figure 2 in this section.

The primary outcome (mortality) is not clear. Recommended to include the biostatistical equation for mortality.

Line 206: What about predictors for incidence?

Did the authors performed a regression analysis adjusted to potential confounders?

5. RESULTS

Line 212: The abstract mentions 56 patients with HD, and 20 with PD. Please revise.

Lines 214, 215, 216: Please include denominators for sample size (n/N, %).

Primary outcome: Please utilize denominators in this paragraph as well. Again, line 231 refers to general population in Qatar regardless of dialysis.

Recommended to report significant associations with mortality from table 2 as text.

Line 241-244: Report p-value and 95% CI next to OR.

Secondary outcomes: Line 257-260 please elaborate.

Line 262: include objective data for total dialysis population.

Recommended to include association of risk factors with incidence of covid-19 in patients with dialysis.

Severity and complications: Please include denominators (n/N) next to %

What was the mean ICU length of stay in those 19 patients admitted?

Comparison of PD vs HD: line 287, 289: it cannot be claimed that incidence is lower since difference was not significant.

Line 292: What do the authors mean by comparable?

Line 292, 293, 295 and 296: Please include objective data.

6. DISCUSSION

Line 312-316 should be moved to the introduction section.

321: general population with or without dialysis?

331: Please substitute “ten times greater” by “showed a trend of” since p value was not significant.

335: Please describe the socioeconomic status from reference 13.

348: Please include the x-ray previous reports from reference 18.

352: Peak levels for blood workout in HD vs baseline levels in PD is not a fair comparison. In addition, it is not recommended to state “higher peaks” if difference was not significant.

355: DVT and AF were significantly associated with mortality in chi2 test, but regression analysis was not performed nor reported. Therefore, if the authors use chi2 to report association with mortality, then all chi2 values should also be included. This means chest x ray, and all significant laboratory findings. Recommended to perform regression analysis for all variables.

357: The only significant differences of peak levels for non-survivors were CRP, ferritin, il-6, and WBC count, the rest of the inflammatory markers were not higher. In addition, peak levels for CK, WBC, D-dimer, liver enzymes were not reported. Where is this information coming from? Please revise.

363: Case fatality rate should be reported in the results section. This data was not described before.

7. TABLES

Table 1: The article compares patients under dialysis with and without covid-19 throughout the manuscript (incidence). It is recommended to compare both populations in table 1, including baseline laboratory findings (bloodwork).

Table 2 (Mortality): Recommended to describe those significant associations in regression analysis in the Results section. Please be specific for regression model or chi2. This creates confusion.

Table 3: Recommended to remove table 3 and report complications as text in the Results section. Moreover, it is recommended to illustrate in a figure those laboratory findings that were significantly different (t-test) between baseline and 7 days.

8. FIGURES

Figure 1. Please use a standard flow chart for retrospective studies. Recommended to remove inclusion and exclusion criteria (this should be explained as text). Recommended to include the number (n=971) of the first filter of excluded patients with reason of exclusion. Same for those excluded after “assessed for eligibility”.

The authors provide a good plan to associate mortality of covid-19 dialysis patients with risk factors and laboratory work. This study requires more clarity on weather association was assessed with chi2 or regression analysis. In addition, potential confounders in baseline characteristics between dialysis patients with or without covid-19 (as suggested table 1) should be considered within the model. Lastly, those significant associations with mortality and incidence (after revision) in regression analysis should be further discussed in the correspondent section.

Journal Requirements:

2. Thank you for stating in the text of your manuscript "the study meets the waiver criteria of informed consent, and all data were fully anonymized before we accessed them". Please also add this information to your ethics statement in the online submission form.

3. Thank you for providing the date(s) when patient medical information was initially recorded. Please also include the date(s) on which your research team accessed the databases/records to obtain the retrospective data used in your study."

Reviewers' comments:

Reviewer's Responses to Questions

**Comments to the Author**

1. Is the manuscript technically sound, and do the data support the conclusions?

Reviewer #1: Yes

Reviewer #2: Partly

2. Has the statistical analysis been performed appropriately and rigorously? 

Reviewer #1: Yes

Reviewer #2: No

3. Have the authors made all data underlying the findings in their manuscript fully available?

Reviewer #1: Yes

Reviewer #2: No

4. Is the manuscript presented in an intelligible fashion and written in standard English?

Reviewer #1: Yes

Reviewer #2: Yes

5. Review Comments to the Author

Reviewer #1: The manuscript was well written and highlighted the risk of COVID-19 in people undergoing maintenance dialysis. The results of this manuscript can facilitate in taking appropriate measures to minimize the risk of COVID-19 infection in high risk population. However, authors should address for typo in the abstract and inconsistent fonts within the text before submitting.

Reviewer #2: The authors provide a good plan to associate mortality of covid-19 dialysis patients with risk factors and laboratory work. This study requires more clarity on weather association was assessed with chi2 or regression analysis. In addition, potential confounders in baseline characteristics between dialysis patients with or without covid-19 (as suggested table 1) should be considered within the model. Lastly, those significant associations with mortality and incidence (after revision) in regression analysis should be further discussed in the correspondent section.

6. PLOS authors have the option to publish the peer review history of their article (what does this mean?). If published, this will include your full peer review and any attached files.

Reviewer #1: **Yes: **Ram Kinker Mishra

Reviewer #2: No

---

## [Author Response · Author response to Decision Letter 0]

1 May 2021

Dear Professor Bijan:

I like to thank you and the respected reviewers for your hard work to review our manuscript. We tried to answer and implement all valuable comments made by the reviewers. 

I like to emphasize on few major comments made by the reviewers:

1- regarding statistical analysis: it was redone and details as requested. 

2- regarding general population: we clarified that general population in text meant people of Qatar and removed it from the result part. We mentioned it in the discussion to highlight differences in Qatar of general population to dialysis patients (our study cohort).

3- We added control group (dialysis patients not infected with COVID19) as suggested and compared risk factors as suggested by reviewers.

4- Ethics approval 

Ethical approval for the conduct of the study was obtained from Institutional review Board (IRB) of Medical Research Center (MRC) of Hamad Medical Corporation (HMC) (approval Number: MRC-05-161) On 20 July 2020. All procedures performed in the study were in accordance with the good clinical practice and comparable ethical standards.

- All comments has been addressed within the revised manuscript with track changes.

Thank you for considering our manuscript for publication in your esteemed journal.

---

## [Decision Letter · Decision Letter 1]

9 Jun 2021

PONE-D-21-00180R1

Mortality and associated risk factors of COVID-19 infection in dialysis patients in Qatar: a nationwide cohort study

PLOS ONE

Dear Dr. Hamad,

Thank you for submitting your manuscript to PLOS ONE. After careful consideration, we feel that it has merit but does not fully meet PLOS ONE’s publication criteria as it currently stands. Therefore, we invite you to submit a revised version of the manuscript that addresses the points raised during the review process.

We look forward to receiving your revised manuscript.

Kind regards,

Bijan Najafi

Academic Editor

PLOS ONE

Journal Requirements:

Additional Editor Comments (if provided):

Thanks for your efforts in addressing the initial concerns voiced by the reviewers. One of the reviewers raised additional concerns which are valid and should be addressed before I could recommend the acceptance of your manuscript. I however evaluate these concerns to be minor.

Reviewers' comments:

Reviewer's Responses to Questions

**Comments to the Author**

1. If the authors have adequately addressed your comments raised in a previous round of review and you feel that this manuscript is now acceptable for publication, you may indicate that here to bypass the “Comments to the Author” section, enter your conflict of interest statement in the “Confidential to Editor” section, and submit your "Accept" recommendation.

Reviewer #1: All comments have been addressed

Reviewer #2: (No Response)

2. Is the manuscript technically sound, and do the data support the conclusions?

Reviewer #1: Yes

Reviewer #2: Partly

3. Has the statistical analysis been performed appropriately and rigorously? 

Reviewer #1: Yes

Reviewer #2: No

4. Have the authors made all data underlying the findings in their manuscript fully available?

Reviewer #1: Yes

Reviewer #2: (No Response)

5. Is the manuscript presented in an intelligible fashion and written in standard English?

Reviewer #1: Yes

Reviewer #2: Yes

6. Review Comments to the Author

Reviewer #1: Authors have addressed my comments adequately. However, I would recommend to chek for inconsistent fonts within the manuscript.

Reviewer #2: This article is well written. The authors made a great effort to address the previous comments of the reviewers. General population and control group for incidence concerns have been addressed. There is an intention to describe mortality and associated risks. Incidence was reported, however associated risks for incidence were not analyzed with regression, thus, any statement related to associated risks for incidence should be avoided throughout the manuscript. Additionally, the regression analysis for mortality is still hesitant for potential confounders.

Abstract

The abstract should follow a proper sequence. Title says “mortality and associated risks”, but the context says “incidence and outcomes”; these are not the same. Methods is not describing the primary (mortality and associated risk factors [page 19]) and secondary (a, b, and c) outcomes; this creates confusion on what the results are describing. In the results: “Male patients had double risk for contacting COVID-19”. This is an associated risk for incidence, not mortality. Do the authors mean risk for death? The authors should include the associated risk factors for mortality described in page 19/table 2A. The conclusion should state “higher incidence compared to the general population of Qatar” since the 7.1% is higher than prior reports (4%). Same for mortality (15% vs 0.15%). Recommended to clarify these “high” numbers are compared to nationwide.

Body of manuscript

Statistical analysis

The survivor vs non-survivor cohorts have vast significant differences/trends for baseline values (Page 18 and 19/Table 2). Did the cox regression analysis include those as potential confounders for associated risk factors for mortality? Did the authors perform a selection method (i.e., forward, backward, stepwise) to determine truly potential confounders from table 2?

Results

Denominator in line 221 and the flow chart is 1064. The abstract and other sections have 1068, please revise.

For secondary outcome C (line 380), recommended to describe only significant values within the text to avoid confusion.

Discussion

Line 468: Recommended to avoid using “some association”. Whether there is an association or not.

470: “Mortality was more associated with older age and dialysis”. What do the authors mean as “more”. Again, associations should be determined or not. Are these results described previously?

474: Recommended to report the percentages for case fatality rate of Spain, Italy, and Turkey.

Figure 1: I believe there is a typo on the screening box. “tested negative” should be “positive”??

Figure 3: Recommended to use panel labels for each graph (A, B, C, etc). Error bars for albumin and lymphocytes show non-significant difference. Please revise.

7. PLOS authors have the option to publish the peer review history of their article (what does this mean?). If published, this will include your full peer review and any attached files.

Reviewer #1: No

Reviewer #2: No

---

## [Author Response · Author response to Decision Letter 1]

22 Jun 2021

PONE-D-21-00180R1

Mortality and associated risk factors of COVID-19 infection in dialysis patients in Qatar: a nationwide cohort study

PLOS ONE

Dear Editor in Chief and Esteemed Reviewers:

Thank you for your valuable review and feedback

please find our rebuttal letter that responds to each point raised 'Response to Reviewers'.

I want to highlight that we had help from a statistician this time (we added acknowledgment) to fulfill the valuable recommendations of reviewers. 

Journal Requirements:

Please review your reference list to ensure that it is complete and correct. 

We reviewed and updated all references as per journal guidelines. We checked and found no reference has been retracted

I would recommend to check for inconsistent fonts within the manuscript.

thank you for your valuable comment. We made changes necessary as recommended.

Abstract

The abstract should follow a proper sequence. Title says “mortality and associated risks”, but the context says “incidence and outcomes”; these are not the same. 

Thank you for this valued observation. We agree with comment. text changed as recommended.

Methods is not describing the primary (mortality and associated risk factors [page 19]) and secondary (a, b, and c) outcomes; this creates confusion on what the results are describing. 

Agree with remark. Primary and secondary objectives added to methods

In the results: “Male patients had double risk for contacting COVID-19”. This is an associated risk for incidence, not mortality. Do the authors mean risk for death?

Thank you for this comment. We rephrased to double the incidence to make it clear and not related to mortality and avoid any confusion.

 The authors should include the associated risk factors for mortality described in page 19/table 2A. 

Agree with your valuable observation. We added the most important significant risk factors from Table 2A (now 3A) as advised. Also added result from new table 3B (multivariate Cox regression analysis).

The conclusion should state “higher incidence compared to the general population of Qatar” since the 7.1% is higher than prior reports (4%). Same for mortality (15% vs 0.15%). Recommended to clarify these “high” numbers are compared to nationwide.

Thank you for your comment. We added and rephrased the conclusion in the abstract as suggested.

Body of manuscript

Statistical analysis

The survivor vs non-survivor cohorts have vast significant differences/trends for baseline values (Page 18 and 19/Table 2). Did the cox regression analysis include those as potential confounders for associated risk factors for mortality? Did the authors perform a selection method (i.e., forward, backward, stepwise) to determine truly potential confounders from table 2?

Thank you for your valuable comments. The cox regression analysis summarized in table 2A (now changed to 3A) was a univariate analysis and did not account for potential cofounders. Therefore, we constructed in this revision a forward stepwise multivariate cox regression analysis model to further examine the association between the different statistically significant variables in the univariate analysis (Table 2A, p < 0.05) and mortality in dialysis patients infected with COVID-19. The model initially included clinical variables that were statistically significant (p<0.05) and that were thought to be clinically relevant and not collinear. ICU admission was found to be the only clinical variable significantly associated with death [OR = 32.8 (3.5-305.4), p=0.002)] (Table 3B). Then we tried to add the blood investigation variables that were statistically significant in the univariate analysis (Table 2A, p<0.05) to the model. However, the model became unstable due to our small sample size as well as the collinearity between ICU admission and inflammatory and thrombotic blood markers such as lactate, CRP, IL-6, D-dimer and ferritin.

Results

Denominator in line 221 and the flow chart is 1064. The abstract and other sections have 1068, please revise.

Agree and we commend you for sharp observation and we apologize for the typo. all corrected to the correct number 1064.

For secondary outcome C (line 380), recommended to describe only significant values within the text to avoid confusion.

Thank you for the comment. We summarized (very brief) non-significant difference in one sentence and kept the significant ones as advised. For incidence and mortality, we kept it to highlight these important factors though it was not statistically significant.

Discussion

Line 468: Recommended to avoid using “some association”. Whether there is an association or not.

Agree with comment. We rephrased to association with multiple inflammatory markers

470: “Mortality was more associated with older age and dialysis”. What do the authors mean as “more”. Again, associations should be determined or not. Are these results described previously?

Appreciate this observation. We removed more as advised.

474: Recommended to report the percentages for case fatality rate of Spain, Italy, and Turkey.

Done as recommended (as mortality rate in Turkey study was lower than ours, we refrased the sentence to lower than most international figures.

Figure 1: I believe there is a typo on the screening box. “tested negative” should be “positive”??

Yes and again thank you for your close observation and sorry for the typo. Changes to positive.

Figure 3: Recommended to use panel labels for each graph (A, B, C, etc). 

Graph labeled as recommended.

Error bars for albumin and lymphocytes show non-significant difference. Please revise

Agree with valuable comment. After consulting with our statistician and reanalysis, we adjusted the graph

---

## [Editor Report · Decision Letter 2]

24 Jun 2021

Mortality and associated risk factors of COVID-19 infection in dialysis patients in Qatar: a nationwide cohort study

PONE-D-21-00180R2

Dear Dr. Hamad,

We’re pleased to inform you that your manuscript has been judged scientifically suitable for publication and will be formally accepted for publication once it meets all outstanding technical requirements.

Kind regards,

Bijan Najafi

Academic Editor

PLOS ONE

Additional Editor Comments (optional):

Thank you for your efforts in addressing the remaining concerns voiced by the reviewers. After reviewing the latest version and your response letter I believe your revision is responsive to all remaining critiques and your study has sufficient scientific merit, high novelty, and high significance in furthering our understanding about the impact of COVID-19 on patients receiving HD process. Thus I recommend acceptance of your latest revision of your manuscript in the current form. Congratulation!
---

## [Editor Report · Acceptance letter]

1 Jul 2021

PONE-D-21-00180R2 

Mortality and associated risk factors of COVID-19 infection in dialysis patients in Qatar: a nationwide cohort study 

Dear Dr. Hamad:

I'm pleased to inform you that your manuscript has been deemed suitable for publication in PLOS ONE. Congratulations! Your manuscript is now with our production department. 

Kind regards, 

on behalf of

Dr. Bijan Najafi 

Academic Editor

PLOS ONE